# Post-Acute Sequelae of SARS-CoV-2 Infection (PASC) for Patients—3-Year Follow-Up of Patients with Chronic Kidney Disease

**DOI:** 10.3390/biomedicines12061259

**Published:** 2024-06-05

**Authors:** Rumen Filev, Mila Lyubomirova, Boris Bogov, Krassimir Kalinov, Julieta Hristova, Dobrin Svinarov, Alexander Garev, Lionel Rostaing

**Affiliations:** 1Department of Nephrology, Internal Disease Clinic, University Hospital “Saint Anna”, 1750 Sofia, Bulgaria; mljubomirova@yahoo.com (M.L.); bbogov@yahoo.com (B.B.); 2Faculty of Medicine, Medical University Sofia, 1504 Sofia, Bulgaria; julieta_sd@yahoo.com (J.H.); dsvinarov@yahoo.com (D.S.); leksgarev@abv.bg (A.G.); 3Head Biometrics Group, Comac-Medical Ltd., 1404 Sofia, Bulgaria; kkalinov@medistat-bg.com; 4Department of Clinical Laboratory, University Hospital “Alexandrovska”, 1431 Sofia, Bulgaria; 5Cardiology Department, University Hospital “Alexandrovska”, 1431 Sofia, Bulgaria; 6Nephrology, Hemodialysis, Apheresis and Kidney Transplantation Department, Grenoble University Hospital, 38043 Grenoble, France; lrostaing@chu-grenoble.fr; 7Internal Disease Department, Grenoble Alpes University, 38043 Grenoble, France

**Keywords:** long COVID, PASC, comorbidity, pain, dyspnea, CKD

## Abstract

Post-acute sequelae of SARS-CoV-2 (PASC) is a significant health concern, particularly for patients with chronic kidney disease (CKD). This study investigates the long-term outcomes of individuals with CKD who were infected with COVID-19, focusing on their health status over a three-year period post-infection. Data were collected from both CKD and non-CKD patients who survived SARS-CoV-2 infection and were followed for three years as part of a research study on the impact, prognosis, and consequences of COVID-19 infection in CKD patients. In this prospective cohort study, we analyzed clinical records, laboratory findings, and patient-reported outcomes assessed at intervals during follow-up. The results indicated no permanent changes in renal function in any of the groups analyzed, although patients without CKD exhibited faster recovery over time. Furthermore, we examined the effect of RAAS-blocker therapy over time, finding no influence on PASC symptoms or renal function recovery. Regarding PASC symptoms, most patients recovered within a short period, but some required prolonged follow-up and specialized post-recovery management. Following up with patients in the post-COVID-19 period is crucial, as there is still insufficient information and evidence regarding the long-term effects, particularly in relation to CKD.

## 1. Introduction

The COVID-19 pandemic, caused by the novel coronavirus SARS-CoV-2, has forever altered the global health and economic landscape. While significant attention has been focused on the acute phase of the disease, it has become evident that there are long-term health effects in the post-infection period. This phenomenon, characterized by persistent symptoms for months, years, or even longer after the acute infectious phase, continues to challenge healthcare systems.

Long COVID-19 syndrome, also known as post-acute sequelae of SARS-CoV-2 infection (PASC), may have significant implications for individuals with chronic kidney disease (CKD). CKD can be a serious risk factor for complications, as these patients often have compromised immune systems and other underlying health problems. During the acute phase, a common complication is acute kidney injury (AKI). Numerous studies have highlighted that approximately 1 in 10 hospitalized COVID-19 patients experience severe impairment of kidney function [1]. Furthermore, AKI has been identified as a serious risk factor for outcomes, although the incidence and reported outcomes vary widely [1,2]. For example, Hirsch et al. reported a high frequency of AKI (36.6%) in hospitalized COVID-19 patients [2]. Among patients with serious comorbidities, such as cardiovascular disease, hypertension, and diabetes, it is believed that one out of five hospitalized patients will deteriorate to AKI, and around 10% of these patients will require renal replacement therapy (RRT) [3]. This evidence demonstrates that CKD is a significant risk factor for a more severe course of the disease and can worsen the prognosis in some cases. The global prevalence of chronic kidney disease is very high [4].

The COVID-19 pandemic has had a major impact on patients with CKD. AKI is the most common complication of COVID-19 and increases mortality in infected patients, particularly those in intensive care units [5,6]. While most patients who survive COVID-19-associated AKI regain kidney function, up to 30% may remain on dialysis at discharge [7]. The high mortality rate, predisposition to additional complications, and uncertainty about the long-term impact on renal function necessitate serious and thorough follow-up for patients with CKD who have survived COVID-19 infection. A detailed assessment of renal outcomes in the long COVID-19 period is not yet available, and there remains a lack of sufficient information.

PASC is one of the most distressing manifestations reported, characterized by persistent symptoms that can develop within a few weeks after infection and last for months [8]. Despite the severely distressing nature of PASC, long-lasting symptoms can occur in patients who did not experience severe infection or have any comorbidities before the infection [8,9]. Several studies have reported that the most common persistent symptoms after severe COVID-19 infection include fatigue (with or without physical activity), dyspnea, chest pain, cognitive dysfunction, and, in some cases, gastrointestinal problems [8,10,11].

Concerns about PASC syndrome have been raised by reports of increased hospitalizations due to persistent symptoms after surviving COVID-19 infection. In this study, we included patients who were actively followed and who initially participated in a study by our research team to determine early clinical laboratory and prognostic biomarkers for COVID-19 infection outcomes. All participants were successfully cured and were the surviving patients in our cohort at that time. The aim of this study is to determine the risk factors for persistent symptoms of PASC. For all participants, infection was confirmed by a positive PCR test for SARS-CoV-2, following the standardized sample collection protocols [12].

## 2. Materials and Methods

This single-center study was conducted at Saint Anna Hospital in Sofia, Bulgaria. Patient follow-up occurred between 1 September 2022 and 1 February 2024. All patients tested positive for COVID-19 between 1 February and 31 March 2021. Throughout the follow-up period, patients reported having only one infection. After their recovery, they became more cautious, consistently following infection prevention guidelines and using personal protective equipment, although they chose not to be vaccinated against COVID-19. They did not receive any special treatment that could have influenced the outcomes.

All patients monitored in this study had participated in previous research on biomarkers and COVID-19, having survived the viral infection. They were all hospitalized following a positive PCR test for SARS-CoV-2, with none being outpatients. Only patients older than 18 years without urinary tract infections, as confirmed by microbiology tests and urine sediment, were included. To be classified in the CKD group, patients had to have a known medical history of CKD for at least six months before the COVID-19 infection, documented by kidney ultrasound, serum creatinine levels, and 24 h proteinuria or albumin/creatinine ratio (uACR). Patients without prior CKD diagnosis or documentation were excluded. All participants were unvaccinated against COVID-19 and confirmed their unvaccinated status during follow-up. The cohort consisted exclusively of Caucasian patients. Initially, 160 patients were included: 120 tested positive for COVID-19, and 40 served as controls. Among the controls, 20 had CKD (following the same criteria) but tested negative for COVID-19, while the remaining 20 had no comorbidities and also tested negative for COVID-19. Of the COVID-19 positive group, 70 had CKD (none on hemodialysis), and 50 had no history of kidney disease with normal serum creatinine levels (44–80 μmol/L for women and 62–106 μmol/L for men). The estimated glomerular filtration rate (eGFR) was calculated using the CKD-EPI 2021 formula, and all patients were staged according to KDIGO criteria. Among the hospitalized COVID-19 patients, 23 (21 with CKD) did not survive the infection, resulting in a 14.7% mortality rate. All deceased patients were in the ICU and died of progressive and complicated pneumonia, with no de novo vascular incidents. No patients died during the three-year follow-up period.

During the follow-up phase, each patient had one telephone call and two clinic visits for physical and laboratory examinations. Patients completed a symptom questionnaire (provided in Bulgarian and English in Appendix A) at each of the three follow-up points: the 20th month post-positive test (clinic visit), the 28th month (telephone call), and the 33rd–36th month (clinic visit, with timing extended due to technical reasons). Physical examinations and laboratory tests included blood count, creatinine measurement, eGFR calculation using CKD-EPI 2021, uACR measurement, and kidney ultrasound.

The study adhered to the guidelines of the Declaration of Helsinki and received approval from the KENIMUS Ethics Committee at the Medical University of Sofia, Bulgaria, under protocol No. 12/31.05.2022. All patients provided informed consent and none withdrew from the follow-up program.

This single-center study following CKD patients for three years to examine post-acute sequelae of SARS-CoV-2 infection (PASC) has several strengths and weaknesses. The focus on a specific population allows for a detailed assessment of PASC effects within that group. The single-center nature ensures consistency in data collection and completeness of longitudinal data. Given the paucity of long-term COVID-19 data, following patients for a minimum of three years is crucial. However, single-center studies may have limitations in generalizability and sample size, and the results may be influenced by biases related to the specific population served. Currently, there are no collaborative arrangements with other centers, which could enhance the study’s reliability.

### Statistical Analysis

Statistical analysis was performed using SAS^®^ version 9.4. The Shapiro–Wilk test was employed to assess the normality of the distribution of creatinine and ACR. The Wilcoxon signed-rank test was used to determine if changes from baseline were significant.

Associations between variables were evaluated using the point biserial correlation coefficient. The odds ratio (OR) was calculated to quantify the association between ACEI/ARB usage and fatigue, cognitive impairment, pain, and sleep problems. A logistic regression model was used to estimate ORs and their significance. A forest plot was used to graphically present the ORs.

Differences between CKD and non-CKD groups, as well as gender differences in follow-up creatinine and ACR, were examined using ANCOVA. The model included follow-up creatinine or ACR as the dependent variable, with the CKD group and gender as factors, and baseline creatinine or ACR and age as covariates.

All comparisons were based on Least Square Means (LSMeans), adjusted for the aforementioned factors and covariates. Differences between LSMeans were graphically presented using a mean–mean scatter plot. A *p*-value below 0.05 was considered statistically significant.

## 3. Results

Among the 120 patients confirmed positive for COVID-19, 58.3% (70 patients) had a history of CKD, while the remaining 41.7% (50 patients) did not (Table 1). The overall mortality rate was 14.7% (23 patients), with 19 (82.6%) of the deceased having a history of CKD. In the surviving group of 97 patients, 51 (52.5%) had CKD, while the remaining 46 (47.5%) did not.

During the follow-up stage, the median age of CKD patients who tested positive for COVID-19 was 52.6 years, compared to 63.8 years for non-CKD COVID-19 patients (*p* = 0.07). Gender distribution differed between groups, with a total of 43 (44.3%) male patients and 54 (55.6%) female patients (*p* = 0.03). Further analysis revealed that 34 (66.6%) CKD patients were female, while 20 (43.4%) non-CKD patients were female (*p* = 0.03).

During the follow-up, a comparison was conducted between baseline creatinine levels, calculated eGFR, and measured urine albumin/creatinine ratio using the same laboratory tests to evaluate the impact of COVID-19 infection on renal function.

Initially, at baseline, the mean serum creatinine level upon admission was 119.0 μmol/L (range: 50.0–769.0 μmol/L) for the CKD group and 79.0 μmol/L (range: 50.0–295.0 μmol/L) for the non-CKD group (Table 2).

Subsequently, during follow-up, it was observed that patients who survived COVID-19 infection had regained kidney function in terms of serum creatinine levels when comparing the creatinine levels at admission for hospital treatment for COVID-19 infection three years ago to those at the last follow-up. However, no significant differences were noted between the CKD and non-CKD groups regarding renal function recovery (serum creatinine levels), suggesting no substantial distinction in terms of kidney function recovery.

Subsequently, another analysis was conducted comparing all patients at baseline upon admission for hospital treatment with COVID-19 and the follow-up to identify any significant differences regarding creatinine levels. Significant changes were observed in creatinine levels at the last follow-up, with a mean decrease of −4.39 μmol/L (*p*-value < 0.0001) compared to baseline. However, despite the decrease in creatinine levels, the estimated GFR was found to be lower than at hospital admission for COVID-19 infection three years ago, likely influenced by age as a factor affecting eGFR calculation. Additionally, a statistically significant difference was noted in uACR calculations, with a decrease in measured values by a mean difference of −0.47 g/L (*p*-value < 0.0001) when comparing overall results at follow-up with those at hospital admission three years ago.

Among the surviving patients, 22 (22.68%) experienced AKI during the COVID-19 infection, classified by KDIGO criteria: Stage I for 13 patients (13.40%), Stage II for 3 patients (3.09%), and Stage III for 6 patients (6.19%). All patients successfully recovered their kidney function, with a median serum creatinine of 113.5 μmol/L (range: 55.00–200.0 μmol/L), and none required renal replacement therapy during or after the COVID-19 infection.

To obtain more precise results, detailed analyses were performed using ANCOVA to analyze the non-CKD and CKD groups. This method was selected as a useful statistical technique for comparing group means while considering the influence of a continuous covariate, allowing for a better understanding of the relationship between groups and dependent variables while controlling for other relevant factors. An analysis was conducted for the follow-up of creatinine and uACR, with the statistical model adjusted for sex and age. Several comparisons were made for creatinine using GLM (Generalized Linear Models) with computation of Least Square Means (LSMEANS) within the context of the GLM results.

When analyzed by this method, similar results were obtained: creatinine significantly improved three years after the infection (*p*-value < 0.0001). However, differences were observed between the CKD and non-CKD groups, with patients without CKD showing better recovery of renal function than those with any stage of CKD before the COVID-19 infection. Furthermore, females exhibited better recovery of renal function after surviving the COVID-19 infection, while age was found to be non-significant for the results (Table 3 and Figure 1).

Similar analyses were conducted for uACR, yielding comparable results to those for serum creatinine. Significant improvement in uACR was observed three years later, with better improvement noted for non-CKD patients and females exhibiting lower uACR compared to males. Age, again, was found to be non-significant (Table 4 and Table 5 and Figure 2).

The initial follow-up protocols encompassed not only the analysis of laboratory findings but also the comprehensive examination of patients to identify symptoms associated with post-acute sequelae of SARS-CoV-2 (PASC). It is widely acknowledged that symptomatic presentations can vary considerably; however, the symptoms reported by our patients predominantly included fatigue, sleep disturbances, cognitive dysfunction, and notably, a significant proportion of patients reported chest pain or discomfort (Table 6). All symptoms were clearly delineated in the questionnaire, and patients who provided positive responses underwent thorough evaluation. Those requiring further assessment were referred to relevant specialists; for instance, patients reporting chest pain were consulted with a cardiologist and underwent echocardiography ultrasound examinations.

All patients reported that symptoms emerged within a few weeks following the infection, with some enduring for several months (66% of patients), while others persisted for over a year (34% of patients). Subsequent analysis aimed to establish any correlation between the laboratory results and reported post-acute sequelae of SARS-CoV-2 (PASC) symptoms, yet no significant findings or correlations were identified (refer to Table 7). It is noteworthy that no significant differences were observed between the CKD and non-CKD groups, nor between genders, upon comparison. The shortest reported symptom duration was chest pain, lasting up to 6 months, whereas cognitive dysfunction was cited as the most prolonged complaint, enduring for up to 18 months.

Further analysis was conducted concerning reported chest pain, which raised concern for our team due to approximately 35% of patients reporting it in our survey. Patients reporting this symptom underwent additional examinations by pulmonologists and cardiologists. Most patients reported experiencing this symptom for a shorter duration, potentially attributable to reasons such as prolonged coughing or reduced physical activity during COVID-19 recovery, leading to muscle strain or rib discomfort resulting in chest pain. Imaging examinations were conducted to rule out post-COVID inflammation as a cause. In total, 23 patients were assessed by cardiologists, undergoing physical examinations and echocardiography, with five patients further undergoing coronary angiography; however, no additional pathological findings were detected, and no coronary interventions were performed in any cases. With more serious pathological causes for the pain ruled out, patients were incorporated into rehabilitation programs or advised to commence daily physical activity, yielding positive outcomes, particularly for those also reporting fatigue as a symptom.

More than 60% of patients reported experiencing sleeping disturbances and cognitive dysfunction, with many cases involving both symptoms concurrently. Cognitive dysfunction was the longest-lasting symptom, persisting for up to 18 months. Patients described experiencing “brain fog”, characterized by difficulties in memory, concentration, and overall mental clarity. The precise etiology of this symptom remained elusive, as even patients with a milder course of COVID-19 exhibited prolonged post-acute sequelae of SARS-CoV-2 (PASC) symptoms. Cognitive rehabilitation involving brain exercises, stress management techniques, and improved sleep hygiene was recommended to these patients. Additionally, some patients reported experiencing depression and anxiety, prompting referrals for specialized follow-up and support from a psychologist. After a year and a half, none of the patients reported such symptoms, including elderly individuals who were closely monitored.

In an earlier phase of the study involving COVID-19 patients, a comprehensive analysis was conducted on the effects of RAAS-blocker therapy. It was concluded that patients receiving ACE inhibitors (ACEi) and angiotensin receptor blockers (ARBs) exhibited a significantly lower mortality rate compared to those receiving other classes of antihypertensive drugs [13]. This analysis was prompted by conflicting information from various trials regarding the use of RAAS blockers at the onset of the pandemic. Subsequently, during follow-up, an investigation was undertaken to assess the impact of ACEi/ARB therapy on laboratory results and to explore any potential correlations with reported PASC symptoms within our patient cohort. Among the surviving patients, 32 individuals (32.9%) were receiving daily RAAS-blocker therapy, which had been initiated prior to COVID-19 infection and continued thereafter. A logistic regression model was employed to examine the effects of RAAS blockers on reported PASC symptoms, including fatigue, cognitive dysfunction, chest pain, and sleep disturbances, using odds ratio analysis. No significant association between symptoms and therapy was identified, leading to the conclusion that RAAS-blocker therapy had no discernible effect during the post-infection period (Figure 3). Similarly, additional analyses conducted on the effects of ACEi/ARBs on kidney function and the albumin/creatinine ratio (ACR) failed to establish any correlation.

## 4. Discussion

The long-term health consequences of COVID-19 remain largely unclear at this point. Studies investigating the long-term health effects of COVID-19 on patients, regardless of hospitalization status, are ongoing. These studies aim to identify the associated risk factors and determine whether they contribute to a more severe course of the disease. While the cohort of surviving COVID-19 patients who were followed up with did not show permanent kidney damage or further progression of chronic kidney disease (CKD), it is strongly recommended that these patients continue to be monitored over time. Additionally, post-acute sequelae of SARS-CoV-2 (PASC) symptoms has been observed to impact individuals’ lives, regardless of CKD status or severity of initial COVID-19 symptoms.

Previous reports have indicated that CKD is a predisposing factor for acute kidney injury (AKI), which is a negative prognostic factor for mortality rates [5]. Given the clear and significant data from global analyses, questions have arisen regarding the long-term effects of COVID-19 on kidney function, particularly in CKD patients who may be more vulnerable to complications due to their underlying kidney disease. A study by Benjamin Bowe et al., involving over 1.7 million US veterans, revealed that 30-day survivors of COVID-19 exhibited a higher risk of AKI, a decline in estimated glomerular filtration rate (eGFR), and end-stage kidney disease (ESKD), which subsequently increased the risk of complications for CKD patients [14].

Another important question is whether COVID-19 can induce chronic pathological changes in the kidneys of patients without pre-existing CKD. A retrospective study by Kremec et al. investigated this in 1,008 participants selected from a larger cohort of COVID-19 patients [15]. The study found that surviving patients with mild, moderate, or severe clinical manifestations of COVID-19 did not exhibit an increased risk of kidney outcomes after the acute phase of the disease.

Reports have suggested that COVID-19 patients may experience post-infection complications such as breathing difficulties, heart problems, and pathological changes in kidney structures, potentially indicating lifelong defects in organ structures and functioning. To explore this further, Paidas et al. conducted an in-depth pathohistological investigation using the MHV-1 mouse model of COVID-19 [16]. Their findings indicated long-term recovery in liver and kidney function after infection, but more severe pathological changes in organs such as the brain, lungs, and heart were observed one year post-infection. These findings suggest multi-organ changes with possible implications for long-term prognosis, although the severity of changes in kidney structures may be less pronounced compared to other organs [17,18].

We also found that hospitalized patients with COVID-19 infection exhibit symptoms of post-acute sequelae of SARS-CoV-2 infection (PASC) within the first 2–3 months. Another study reported similar findings among non-hospitalized, PCR-confirmed COVID-19 patients, with one-third of symptomatic participants experiencing persistent symptoms after 4 and 12 weeks [10]. However, there is currently no definitive evidence indicating the duration of PASC symptoms, representing a gap in our knowledge. In our study, all patients reported that their symptoms had completely resolved by the third year. Cross-referencing most publications, it is evident that the most common symptoms persisting for the first 3–6 months post-infection are chest pain, coughing with or without dyspnea, and fatigue [9,19,20,21]. Patients often describe periods of symptom remission over time [22]. Papers by Nalbandian A et al. and Davis HE et al. noted over 200 symptoms indicating multi-organ involvement [8,23]. In our study, patients reported the previously mentioned common post-infectious symptoms: fatigue, cognitive difficulties, chest pain, and sleep disturbances.

Our results closely align with a study conducted by Alkodaymi et al. [24], although their study assessed PASC symptoms in a much larger population and at different time intervals (≥3 to <6, ≥6 to <9, ≥9 to <12, and ≥12 months). The reported results indicate a lower percentage of patients with complaints before the first 6 months, whereas at around the 12-month follow-up, the results became similar to those in our cohort but slightly lower. Tabacof et al. reported cognitive dysfunction in more than half (63%) of the patients analyzed, with one-third experiencing anxiety and depression, similar to our follow-up results [25]. However, it is challenging to draw conclusions about the impact of PASC on anxiety and depression, as the results may be comparable to those in the general population [26]. Notably, the number of patients reporting such psychoneurological symptoms before COVID-19 infection was lower. The pandemic led to a 27.6% increase in cases of major depressive disorders and a 25.6% increase in cases of anxiety disorders globally [27].

A cluster of cardiovascular (CV) abnormalities has been reported in patients in the post-acute phase, including myocardial inflammation, myocardial infarction, right ventricular dysfunction, and arrhythmias. The pathophysiological mechanisms underlying these late cardiac symptoms are not well understood. Numerous reported cases indicate elevated cardiac troponin levels in COVID-19 patients, suggesting myocardial injury and/or ischemia [28], supported by a pathohistological report of microthrombi found in 15 patients [29]. However, the pathological mechanism in the post-COVID phase remains poorly understood. Common to all studies is a slight increase in the percentage of patients with cardiopulmonary symptoms in the first 3 months compared to later stages around 6–9 months [30,31,32,33,34]. Although most PASC cases with cardiopulmonary symptoms do not reveal significant pathological findings requiring intervention, active follow-up, investigation, and diagnostic refinement of these patients are highly recommended. Regarding therapy with RAAS blockers, there are no reported data on the long-term effects on patients post-COVID-19 infection. Current strong evidence supports the benefits of this therapy only during the acute phase of the viral infection, as demonstrated in the early stages of our research on this cohort [13,35].

Additionally, it is important to note that all patients in our cohort were not vaccinated before their COVID-19 infection and did not vaccinate afterward. Some studies have suggested a relationship between long-COVID complications and a full course of vaccination. Watanabe A et al. reported that COVID-19 vaccination before SARS-CoV-2 infection was associated with a lower risk of long COVID after analyzing six observational studies involving 536,291 unvaccinated and 84,603 vaccinated individuals before SARS-CoV-2 infection [36]. Two-dose vaccination was associated with a lower risk of long COVID compared to no vaccination (OR, 0.64; 95% confidence interval [CI], 0.45–0.92) and one-dose vaccination (OR, 0.60; 95% CI, 0.43–0.83). Two-dose vaccination compared to no vaccination was also associated with a lower risk of persistent fatigue (OR, 0.62; 95% CI, 0.41–0.93) and pulmonary disorder (OR, 0.50; 95% CI, 0.47–0.52) [36]. Another study by Gao et al. supported these findings, indicating that COVID-19 vaccines reduce the risk of long COVID. This study highlighted that the protective effect was observed in participants vaccinated with two doses, but not one dose, emphasizing the importance of vaccination before or even after COVID-19 infection, as vaccination was effective against PASC symptoms of long COVID [37]. Notably, during the pandemic, Bulgaria had one of the lowest two-dose vaccination rates and one of the highest mortality rates in the European Union, underscoring the value of vaccination [38].

A significant aspect of PASC symptoms following COVID-19 is the reported differences between hospitalized and non-hospitalized patients. One meta-analysis indicated that the global prevalence of PASC was higher in hospitalized than non-hospitalized patients—54% versus 34% [39]. A meta-analysis by Fernández-de-las-Peñas et al. reported that the most common PASC symptoms among non-hospitalized patients were smell disturbance, taste disturbance, and dyspnea, whereas fatigue was less commonly reported among non-hospitalized patients [40]. Future investigations of PASC should consider patients with varying severity of COVID-19 infection, including those treated in outpatient settings rather than in hospitals. In our cohort, we only included hospitalized patients; thus, further analysis is needed on non-hospitalized patients and the symptoms they reported. However, the data we collected are very similar to that reported for patients who were actively treated in hospitals.

Furthermore, the relationship between PASC and RAAS blockade therapy presents an interesting avenue of research. PASC encompasses a range of persistent symptoms and organ dysfunction following acute COVID-19 infection, and it is intriguing to explore whether there is any relationship between RAAS therapy and the severity of symptoms. Although our research did not find such a relationship, a more in-depth analysis could be considered in the future. The anti-inflammatory effects of RAAS blockers are well known and studied [41], and they could theoretically reduce the chronic inflammation seen in PASC by lowering angiotensin II levels and increasing angiotensin-(1-7). However, the diverse range of PASC symptoms may require more targeted therapies, as neurological symptoms may not respond to RAAS blockers. In the earlier stages of our study, when we analyzed the acute stage of COVID-19, we found a significant effect on the severity of the infection, with patients on ARB/ACEi therapy experiencing fewer transfers to intensive care due to deterioration of their general condition amidst active COVID-19 infection [13]. We believe a thorough analysis should be conducted to determine whether RAAS-blocking therapy has any effect on long-term post-COVID symptoms.

## 5. Conclusions

Patients with CKD are likely to have long-term sequelae of COVID-19 infection. Although this study did not demonstrate any permanent impairment of renal function or progression of CKD, we strongly recommend that patients with proven renal disease who are survivors of COVID-19 infection be followed up in the future, especially males. We found that even without a direct impact on CKD, PASC can significantly affect the lives of COVID-19 survivors.

## Figures and Tables

**Figure 1 biomedicines-12-01259-f001:**
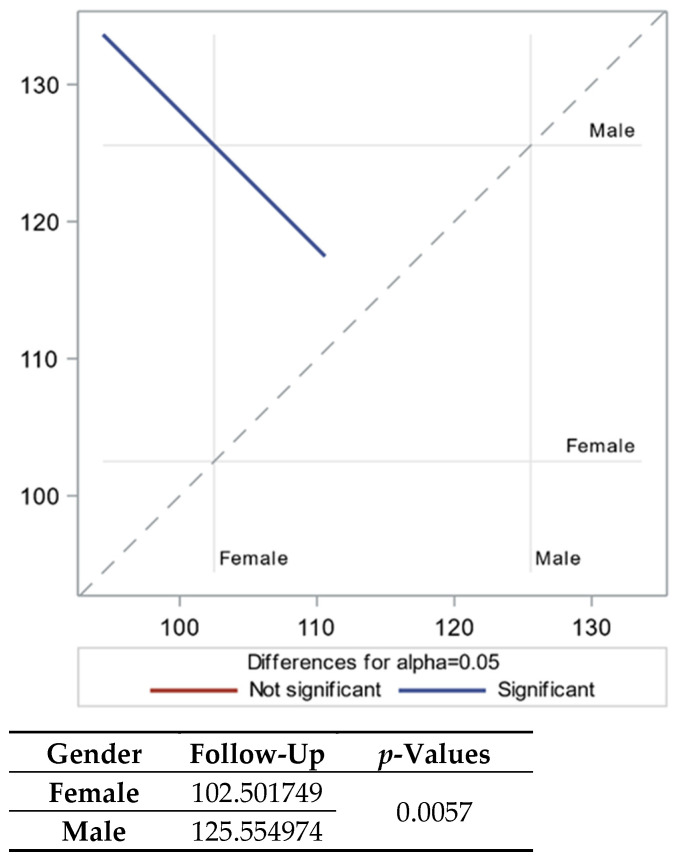
Follow-up creatinine (μmol/L) with LSMEAN for gender.

**Figure 2 biomedicines-12-01259-f002:**
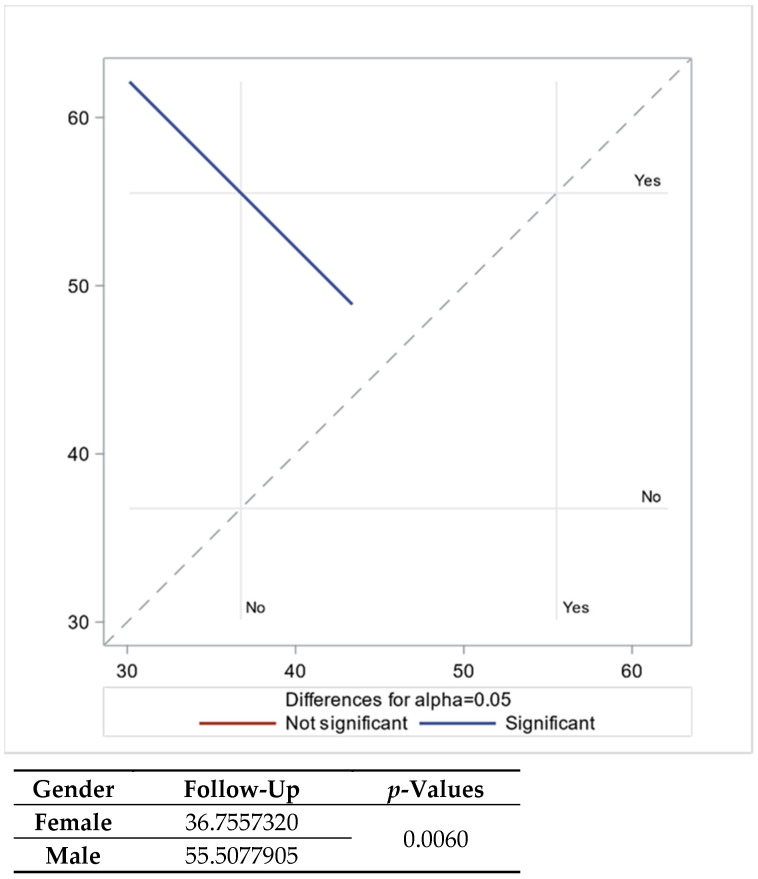
Follow-up uACR with LSMEAN for gender.

**Figure 3 biomedicines-12-01259-f003:**
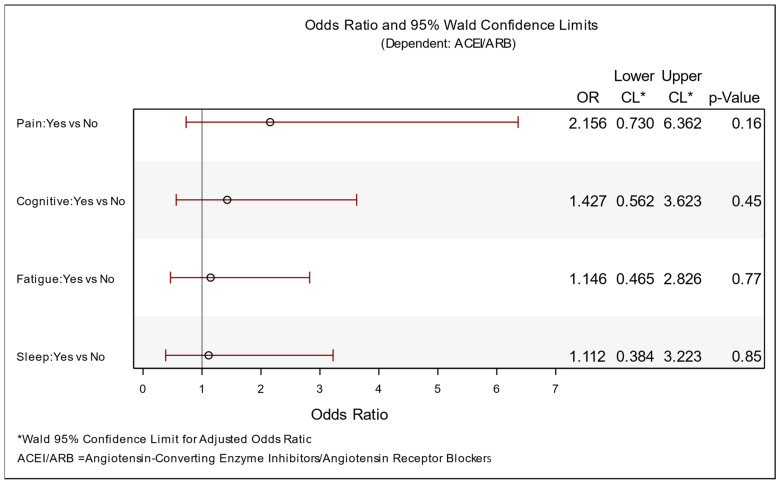
Logistic regression model: odds ratios (dependent: ACEI/ARB).

**Table 1 biomedicines-12-01259-t001:** Analysis of the cohort at the beginning of the study.

Group	N	Gender	Middle Age Mean (SD)	Race (n%)	Arterial Hypertension	Diabetes	CVD
**CKD + COVID-19**	70	Female: 21 (42.0%)	56.8 (16.0)	Caucasians 100%	**Yes** 67 (95.7%)	**Yes** 40 (57.1%)	**Yes** 36 (51.4%)
		Male: 29 (58.0%)			**No** 3 (4.3%)	**No** 30 (42.9%)	**No** 34 (47.6%)
**Non-CKD + COVID-19**	50	Female: 39 (55.7%)	65.9 (12.6)	Caucasians 100%	**Yes** 12 (24%)	**Yes** 1 (2%)	**Yes** 9 (82%)
		Male: 31 (44.3%)			**No** 38 (76%)	**No** 49 (98%)	**No** 41 (18%)
**CKD without COVID-19**	20	Female: 11 (55%)	66.1 (11.8)	Caucasians 100%	**Yes** 18 (90%)	**Yes** 14 (70%)	**Yes** 9 (45%)
		Male: 9 (45%)			**No** 2 (10%)	**No** 6 (30%)	**No** 11 (55%)
**Absolutely healthy**	20	Female: 10 (50%)	36.8 (7.8)	Caucasians 100%	**None of the patients**	**None of the patients**	**None of the patients**
		Male: 10 (50%)					
***p*-values**		0.49	<0.0001	NA	<0.0001	<0.0001	<0.0001

Abbreviations: CKD—chronic kidney disease; SD—standard deviation; CVD—cardiovascular diseases.

**Table 2 biomedicines-12-01259-t002:** Comparison between the groups at the baseline.

*Parameter*	*Statistical Analysis*	*CKD Patients + COVID-19*	*Non-CKD Patients + COVID-19*	*CKD Patients without COVID-19*	*Healthy Controls*	*p-Value*
*Number of patients*	*n*	*70*	*50*	*20*	*20*	
*Creatinine (mcmol/L)*	*Median* *(Ranges)*	*119* *(57.0–930.0)*	*79* *(50.0–1295.0)*	139.1 (86.0–206.0)	60 (49.0–83.0)	*0.0000*
*eGFR (mL/min)*	*Mean* *(SD)*	47.9 (23.0)	80.4 (28.9)	62.3 (22.6)	111.1 (13.0)	<0.0001
*Urea (mmol/L)*	*Median* *(Ranges)*	*9.2* *(2.7–75.2)*	*5.5* *(3.0–82.3)*	*8.0* *(5.0–23.0)*	*3.0* *(2.8–5.0)*	*0.0000*
*D-Dimer (mg/L FEU)*	*Median* *(Ranges)*	*0.9* *(0.3–10.7)*	*0.5* *(0.3–8.1)*	*0.7* *(0.6–4.7)*	*0.2* *(0.1–0.4)*	*0.0000*
*Leucocytes (10^9^ cpl)*	*Mean* *(SD)*	14.8 (11.2)	12.1 (10.5)	8.2 (6.8)	4.5 (2.5)	<0.0001
*CRP (mg/L)*	*Median* *(Ranges)*	*50.8* *(0.5–320.6)*	*29.8* *(0.1–217.6)*	*17.0* *(1.1–30.0)*	*1.5* *(0.1–5.0)*	*0.0003*

Abbreviation: CKD—chronic kidney disease; eGFR—estimated glomerular filtration rate; CRP—C-reactive protein; SD—standard deviation.

**Table 3 biomedicines-12-01259-t003:** Follow-up results for creatinine divided into groups.

Source	DF	Type III SS	MS	*F* Value	*p*-Value
**CKD (Yes/No)**	1	17,817.1	17,817.1	12.10	0.0008
**Gender**	1	11,804.1	11,804.1	8.01	0.0057
**Age**	1	4622.4	4622.4	3.14	0.0798

Abbreviation: CKD—chronic kidney disease; DF—Degrees of Freedom; SS—Sum of Squares; MS—Mean Square; *F* value—Fisher statistics.

**Table 4 biomedicines-12-01259-t004:** Follow-up results for uACR (g/mg).

Source	DF	SS	MS	*F* Value	*p*-Value
**Model**	4	252,699.1	63,174.7	79.72	<0.0001
**Error**	92	72,904.1	792.4		
**Corrected Total**	96	325,603.3			

Abbreviation: DF—Degrees of Freedom; SS—Sum of Squares; MS—Mean Square; *F* value—Fisher statistics.

**Table 5 biomedicines-12-01259-t005:** Follow-up results for uACR divided into groups.

Source	DF	Type III SS	MS	*F* Value	*p*-Value
**CKD (Yes/No)**	1	6266.8	6266.8	7.91	0.0060
**Gender**	1	5358.2	5358.2	6.76	0.0109
**Age**	1	148.7	148.7	0.19	0.6658

Abbreviation: CKD—chronic kidney disease; DF—Degrees of Freedom; SS—Sum of Squares; MS—Mean Square; *F* value—Fisher statistics.

**Table 6 biomedicines-12-01259-t006:** Analysis of the PASC symptoms.

Parameter	Medical Examination	Calculations
**Fatigue** *(Up to 12 months)*	Yes	57 (58.76%)
	No	40 (41.24%)
**Cognitive disfunction** *(Up to 18 months)*	Yes	67 (69.07%)
	No	30 (30.93%)
**Chest pain** *(Up to 6 months)*	Yes	35 (36.08%)
	No	62 (63.92%)
**Sleep disturbances** *(Up to 12 months)*	Yes	60 (61.86%)
	No	37 (38.14%)

**Table 7 biomedicines-12-01259-t007:** Correlation between symptoms and follow-up laboratory results.

Laboratory Results and Symptoms	Creatinine (*p*-Values)	eGFR (*p*-Values)	ACR (*p*-Values)
**Fatigue**	0.4040	0.1506	0.2740
**Cognitive disfunction**	0.1870	0.0600	0.5398
**Chest pain**	0.6632	0.1335	0.3814
**Sleep disturbances**	0.7011	0.2278	0.4360

Abbreviations: eGFR—estimated glomerular filtration rate; ACR—albumin creatinine ratio measured from urine.

## Data Availability

Data are contained within the article or Appendix A.

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
