# Peer review of "Post-Acute Sequelae of SARS-CoV-2 Infection (PASC) for Patients—3-Year Follow-Up of Patients with Chronic Kidney Disease"

_biomedicines, 2024, doi:10.3390/biomedicines12061259_

Round 1

Reviewer 1 Report

Comments and Suggestions for Authors

Thank you for sharing your article on post-acute sequelae of SARS-CoV2 infection among patients suffering from chronic kidney disease followed-up for a 3-year period. The following comments may help to improve the article.

L54-55: Please be more specific in your manuscript regarding "systemic disease".

L56-59: Please revise the sentence for more clarity.

L90: Is the follow-up period stated here correct as you stated a 3-year follow-up in the title? When prior to the follow-up were participants tested positive for COVID-19 infection? Did they have to have a single infection or possible even multiple infections with COVID-19? Did they receive any treatment that could have impacted the outcome assessed within this research?

L94: Participants were excluded if they had possibly an acute urinary tract infection? At what time point within your study and how was it confirmed? How did you confirm the chronic kidney disease? Any chronic kidney disease?  

General comment: Please be more clear on your inclusion and exclusion criteria. 

L95 and 96: Vaccinated against what? 

L98-99: 160 patients were included. 120 of them had a positive PCR for COVID-19 and 23 of them did not survive the infection. Is the remainder the group of 40 patients serving as a control groups as stated in L104-105? The 120 participants were split into a group of 70 and 50, which is not fully clear as you stated that 23 died. Please revise this section for more clarity. How was it confirmed that the 40 controls did neither have COVID-19 infection nor chronic kidney disease? Is this a kind of convenience sample or did you perform a sample size calculation? 

L107: At what time points during your entire follow-up did the phone call and the 2 clinic visits take place? 

L107-108: Please be more specific which physical and laboratory exams were performed. Regarding lab parameters, please include a section describing how they were performed and on which samples.

L113-115: Were participants not consented? 

L282: It seems that the symptoms assessed were self-reported. Did you provide some standardised definitions for the different symptoms to participants to assure the data you collected were sound? 

L135: It is not clear when during your investigations some of the participants died. Please be more specific when revising your manuscript. 

General comment: Please include a section stating the strengths and weaknesses of your study. 

Comments on the Quality of English Language

The manuscript needs English editing for more clarity.

Author Response

We would like to thank reviewer number 1 for taking the time to review our paper and for giving us the opportunity to improve our work. 

The response to the comments is uploaded as a file. 

All changes are made to the original paper - corrected version (with track changes) and clean version. 

Reviewer 2 Report

Comments and Suggestions for Authors

Here are some suggestions aimed at improving the quality of the study, making it more robust and complete, and facilitating the understanding and interpretation of the results.

The description of the methods is somewhat generic. It would be helpful to provide specific details regarding the inclusion and exclusion criteria of the patients and the methodology used for data collection. It would also be useful to provide details on the age distribution of the participants and the gender breakdown; specify whether the visits and laboratory tests performed during the follow-up were standardized and which specific tests were carried out. The description of the 40 control patients is rather vague. It would be useful to know whether the controls were completely healthy or had other non-COVID-19 related conditions. It is suggested to include a table summarizing the demographic and clinical characteristics of the patients at the beginning of the study (e.g., age, gender, comorbidities) to improve clarity. Additionally, a more in-depth analysis of the 23 deceased patients (e.g., age, pre-existing conditions) is necessary, as it could provide further useful information on COVID-19 associated mortality.

Regarding the generalization of the results, the study states that "there are no permanent changes in renal function," but it is unclear if this applies to all CKD patients or only a specific subset. In the results, the analysis of PASC symptoms and their recovery is not thoroughly covered. An in-depth examination of which symptoms persist the most and their severity would enrich the work. In the discussion, the statement that RAAS-blocker therapy does not affect PASC symptoms or renal function recovery could benefit from a deeper analysis of the underlying mechanisms and clinical implications.

Author Response

We would like to thank reviewer number 2 for taking the time to review our paper and for giving us very clear and important ideas and giving us chance to improve our work. 

The response to the comments is uploaded as a file. 

All changes are made to the original paper - corrected version (with track changes) and clean version. 

Round 2

Reviewer 1 Report

Comments and Suggestions for Authors

Thank you for sharing the revised manuscript. All my comments were addressed sufficiently.